# Formation and Electrochemical Evaluation of Polyaniline and Polypyrrole Nanocomposites Based on Glucose Oxidase and Gold Nanostructures

**DOI:** 10.3390/polym12123026

**Published:** 2020-12-17

**Authors:** Natalija German, Almira Ramanaviciene, Arunas Ramanavicius

**Affiliations:** 1Department of Immunology, State Research Institute Centre for Innovative Medicine, Santariskiu 5, LT-08406 Vilnius, Lithuania; natalija.german@imcentras.lt (N.G.); almira.ramanaviciene@chf.vu.lt (A.R.); 2NanoTechnas—Center of Nanotechnology and Materials Science, Faculty of Chemistry and Geosciences, Vilnius University, LT-03225 Vilnius, Lithuania; 3Department of Physical Chemistry, Faculty of Chemistry and Geosciences, Vilnius University, Naugarduko 24, LT-03225 Vilnius, Lithuania; 4Division of Materials Science and Electronics, State Scientific Research Institute Center for Physical Sciences and Technology, Savanorių Ave. 231, LT-02300 Vilnius, Lithuania

**Keywords:** biofuel cell, cyclic voltammetry, glucose biosensor, glucose oxidase, gold nanoparticles, conducting polymers, polyaniline, polymer nanocomposite, polypyrrole

## Abstract

Nanocomposites based on two conducting polymers, polyaniline (PANI) and polypyrrole (Ppy), with embedded glucose oxidase (GOx) and 6 nm size gold nanoparticles (AuNPs_(6nm)_) or gold-nanoclusters formed from chloroaurate ions (AuCl_4_^−^), were synthesized by enzyme-assisted polymerization. Charge (electron) transfer in systems based on PANI/AuNPs_(6nm)_-GOx, PANI/AuNPs_(AuCl4_^−^_)_-GOx, Ppy/AuNPs_(6nm)_-GOx and Ppy/AuNPs_(AuCl4_^−^_)_-GOx nanocomposites was investigated. Cyclic voltammetry (CV)-based investigations showed that the reported polymer nanocomposites are able to facilitate electron transfer from enzyme to the graphite rod (GR) electrode. Significantly higher anodic current and well-defined red-ox peaks were observed at a scan rate of 0.10 V s^−1^. Logarithmic function of anodic current (log *I*_pa_), which was determined by CV-based experiments performed with glucose, was proportional to the logarithmic function of a scan rate (log *v*) in the range of 0.699–2.48 mV s^−1^, and it indicates that diffusion-controlled electrochemical processes were limiting the kinetics of the analytical signal. The most efficient nanocomposite structure for the design of the reported glucose biosensor was based on two-day formed Ppy/AuNPs_(AuCl4_^−^_)_-GOx nanocomposites. GR/Ppy/AuNPs_(AuCl4_^−^_)_-GOx was characterized by the linear dependence of the analytical signal on glucose concentration in the range from 0.1 to 0.70 mmol L^−1^, the sensitivity of 4.31 mA mM cm^−2^, the limit of detection of 0.10 mmol L^−1^ and the half-life period of 19 days.

## 1. Introduction

The development of easy, fast and cheap technology for the determination of glucose is very important in clinical diagnostics, biotechnology, biofuel cells and food manufacturing [1,2,3,4,5,6,7,8,9,10]. Another important branch, which recently appeared in the area of biotechnology, is the development of glucose-powered biofuel cells for possible electric current supply for implantable sensors and other bioelectronics devices [11]. These emerging research directions are requiring new advanced materials that are able to facilitate charge-transfer from red-ox enzymes towards electrodes [12,13]. Electrochemical enzymatic biosensors combine the high specificity of the enzyme (glucose oxidase (GOx), glucose dehydrogenase) and the sensitivity of electrochemical transducers [2,7,8,14,15]. GOx is a homodymeric enzyme with a flavin adenine dinucleotide (FAD) molecule in each out of two active sites. FAD is tightly bound to the GOx-apo-protein subunit and is involved in the catalytic action of GOx and the formation of the electroactive products suitable for both electrochemical glucose sensing [2,14,16] and spontaneous generation of electrical current. The main advantages of electrochemical glucose biosensors are based on high sensitivity, simplicity in application, accuracy and fast analytical response [1,17]. Therefore, the research related to the development of electrodes, discrimination membranes and immobilization methods, the application of new nanomaterials and of new electrochemically active compounds, which can improve the performance of glucose biosensors, is perennially performed [1,2,3,6,16].

Through the achievements in nanotechnology and nanoscience, the amplification of the electrochemical signal of the sensor and efficiency of biofuel cells can be achieved [2,5,15,18,19,20,21,22,23,24]. Metal nanoparticles and nanostructures of gold, silver and cooper are used in recent modifications of glucose biosensors due to their unique catalytic and/or charge transfer properties [21,22,23,25]. The electrochemical performance of graphite-based sensors can be improved by the modification with nanoparticles [3,25,26,27,28] and/or other nanostructures [17,19,20,23,29,30,31]. Among many metal-based nanoparticles, gold nanoparticles (AuNPs) are the most promising nanomaterials due to their large surface area, high loading efficiency, possible involvement in direct and indirect charge transfer and stabilization of the enzyme through electrostatic interactions and/or strong binding with sulfur-containing groups [2,4,18,32]. Dendritic gold nanostructures of 10–200 nm are characterized by three-order hierarchy-based fractals [22,23,33]. Positively electrically charged gold nanostructures can establish stabile interaction with negatively electrically charged aptamers [31]. Gold compounds are used as connectors between the surface of an electrode and the red-ox center of GOx [20]. In situ reduction is the most popular method for the formation of 13–50 nm size AuNPs [5,18,28,34,35,36]. Therefore, the use of colloidal gold nanoparticles and gold nanostructures for the development of electrochemical sensing devices is an extremely promising prospect [5,19,20,23,25,26,27,34,36].

Electrodes modified by conducting polymers have been extensively developed during the last decades [2,10,28,37,38]. Red-ox reactions are able to induce the formation of charge carriers in conducting polymer backbone, which are polarons (radical ions), bipolarons (dications or dianions) and solitons [39]. The conductivity of polymers can be tuned by the variation of dopant concentration within a conducting polymer-based structure and it becomes higher at higher doping levels [39]. Some conducting polymer-based composites containing carbon or gold-based nanostructures are characterized by excellent electrical conductivity and can also serve as electrode material suitable for the fabrication of biosensors [2,3,28,39,40]. In this way, an effective surface-area of electrode is significantly increased and it ensures efficient exposure of a higher number of active sites and faster mass transfer, which leads to advanced activity of the enzyme and, therefore, higher signal of the biosensor [3,4,10,28,39] and/or larger electrical power of a biofuel cell [41] can be achieved.

Different polymer/AuNPs nanocomposites formation methods have been developed: (i) electrochemical synthesis of conducting polymers in the presence of AuNPs, which is based on monomer’s oxidation with following incorporation of metallic nanoparticles [42], (ii) in situ preparation of AuNPs confined directly in the polymeric matrix [27,43,44] and (iii) coating the AuNPs template with a polymeric layer [25,26]. Firstly, Au^3+^ cation is reduced and forms Au^+^ cation, which is related to the nuclei formation; then, Au^+^ cation is reduced to Au^0^ by the formation of gold clusters [20,30,40]. Au^+^ cation is able to activate inert conjugated molecules (including alkenes, 1,3-dienes, etc., [42]) with the following formation of Au^+^-based complexes. Some parts of such oxidized polymer chain are interacting with formed gold clusters and AuNPs [40]. The incorporation of AuNPs within polymers is an important aspect in the synthesis of conductive composites [2,43,44]. Biosensors based on the GOx, AuNPs and modified by a polymeric layer have found promising applications in glucose sensing during clinical analysis [25,26,27], for food processing [27]. Vilian et al. developed a mediator-free glucose biosensor based on glassy carbon electrode (GC) modified by GOx, poly-L-arginine film (P-L-Arg) and functionalized multiwalled carbon nanotubes (f-MWCNT), which was characterized by high sensitivity of 48.86 μA cm^−2^ mM^−1^ [10]. The same research group reported another highly sensitive (11.65 μA mM cm^−2^) glucose biosensor, which was based on a GC electrode modified by GOx-poly(L-lysine)/reduced graphene oxide-ZrO_2_ composites and was operating in the absence of red-ox mediators. Another group of researchers developed a mediator-free fructose sensor based on a GC electrode modified by a composite consisting of 790 nm AuNPs, Ppy, chitosane and fructose dehydrogenase, with high biocatalytic current density of 2.45 mA cm^−2^ [40]. Wang and Zhang developed a mediator-free glucose biosensor based on a GC electrode modified by graphene/nano-Au/GOx, which exhibited sensitivity of 56.93 μA mM^−1^cm^−2^ [19].

The main aim of the present study was to evaluate some characteristics of a glucose biosensor based on polyaniline (PANI) and polypyrrole (Ppy), with embedded GOx and 6 nm size AuNPs or gold-nanoclusters formed from chloroaurate ions (AuCl_4_^−^) (PANI/AuNPs_(6nm)_-GOx, PANI/AuNPs_(AuCl4_^−^_)_-GOx, Ppy/AuNPs_(6nm)_-GOx and Ppy/AuNPs_(AuCl4_^−^_)_-GOx) nanocomposites. Cyclic voltammetry (CV) was applied for the investigation of analytical characteristics of the glucose biosensor and for the evaluation of these modified electrodes in the determination of glucose in real serum samples in the presence/absence of some interfering materials.

## 2. Materials and Methods

### 2.1. Materials

Glucose oxidase (EC 1.1.3.4, type VII, from *Aspergillus niger*, 201 units mg^−1^ protein) was purchased from Fluka (Buchs, Switzerland), tetrachloroauric acid (HAuCl_4_·3H_2_O) – from Alfa Aesar GmbH & Co KG (Karlsruhe, Germany), sodium citrate – from Penta (Praha, Czech Republic), and tannic acid, D-(+)-glucose, D(–)-fructose, D(+)-mannose, D(+)-galactose, D(+)-xylose and D(+)-saccharose – from Carl Roth GmbH + Co (Karlsruhe, Germany). The solutions of sugars were stored for 1 day before the measurements in order to establish equilibrium between α and β isomers of sugars. The buffer of sodium acetate (SA, 0.05 mol L^−1^ CH_3_COONa·3H_2_O) in the presence of 0.1 mol L^−1^ KCl was prepared from sodium acetate trihydrate and potassium chloride, which were obtained from Reanal (Budapest, Hungary) and Lachema (Neratovice, Czech Republic), respectively. Aniline and sodium hydroxide were obtained from Merck KGaA (Darmstadt, Germany), phenazine methosulfate (PMS), L-ascorbic acid (AA) and uric acid (UA) – from AppliChem GmbH (Darmstadt, Germany), pyrrole – from Acros Organics (Morris Plains, NJ, USA), and hydrochloric acid – from Sigma-Aldrich (Saint Louis, MO, USA). The reagents were characterized as analytically pure or the highest quality. Colored components of PANI and Ppy were removed by filtration through a filtration column (of 5 cm length) filled with α-Al_2_O_3_ with 0.3 µm diameter grains (Type N), which was purchased from Electron Microscopy Sciences (Hatfield, MA, USA). All working solutions were prepared using deionized water and were stored between investigations at 4 °C. 6 nm AuNPs of 2.3 × 10^16^ particles L^−1^ were synthesized according to a methodology presented previously [45].

### 2.2. Enzyme-Assisted Formation and Separation of PANI/AuNPs_(6nm)_-GOx, PANI/AuNPs_(AuCl4_^−^_)_-GOx, Ppy/AuNPs_(6nm)_-GOx and Ppy/AuNPs_(AuCl4_^−^_)_-GOx Nanocomposites

PANI/AuNPs_(6nm)_-GOx, PANI/AuNPs_(AuCl4_^−^_)_-GOx, Ppy/AuNPs_(6nm)_-GOx and Ppy/AuNPs_(AuCl4_^−^_)_-GOx nanocomposites were formed by enzyme-assisted synthesis at room temperature (20 ± 2 °C) in the dark over 2 or 4 days. The formation of polymer nanocomposites (PNC) was performed in 0.05 mol L^−1^ SA buffer, pH 6.0, containing 0.05 mol L^−1^ of glucose, 0.75 mg mL^−1^ of GOx, 0.50 mol L^−1^ of aniline or pyrrole in the presence of 0.46 × 10^16^ particles L^−1^ of AuNPs_(6 nm)_ or 0.6 mmol L^−1^ of HAuCl_4_. To separate enzyme-assisted synthesized nanocomposites from the polymerization solution, a centrifugation by centrifuge IEC CL31R Multispeed (ZI Aze Bellitournt, France) during 8 min (14.6 × 10^3^ g) was performed. Then, PNC were washed two times with 0.05 mol L^−1^ SA buffer, pH 6.0, and then collected by the centrifugation and were used for further measurements. PANI/GOx and Ppy/GOx nanocomposites were formed during a synthesis procedure lasting 2 days, according to the above-described protocol in the absence and presence of AuNPs.

### 2.3. The Preparation of Graphite Rod (GR) Electrode Modified by PANI/AuNPs_(6nm)_-GOx, PANI/AuNPs_(AuCl4_^−^_)_-GOx, Ppy/AuNPs_(6nm)_-GOx and Ppy/AuNPs_(AuCl4_^−^_)_-GOx Nanocomposites for Electrochemical Investigations

Graphite rod (surface area 0.071 cm^2^) served as a working electrode and was used for electrochemical investigations. To design the working electrode, GR was cut, polished firstly by fine emery paper and then by slurry α-Al_2_O_3_ powder. After that, GR was washed by deionized water, dried at room temperature and sealed within a silicone tube to avoid any contact of electrode sides with a solution during electrochemical investigations. Enzyme-assisted synthesized PNC were deposited on the surface of the GR electrode and damped by a drop of 0.05 mol L^−1^ SA buffer, pH 6.0. Then, the surface of the working electrode was covered with a polycarbonate membrane, to avoid the detachment of PNC.

### 2.4. Characterization of PANI/AuNPs_(AuCl4_^−^_)_-GOx and Ppy/AuNPs_(AuCl4_^−^_)_-GOx Nanocomposites

Enzyme-assisted formation of PANI/AuNPs_(AuCl4_^−^_)_-GOx and Ppy/AuNPs_(AuCl4_^−^_)_-GOx nanocomposites proceeded within 2 days. PNC were dispersed and homogenized in 50 μL of deionized water. Then, PNC were deposited on the surface of the Si substrate and the morphology and size of this structure were evaluated by a Hitachi SU-70 (Dublin, Ireland) field emission scanning microscope (FE-SEM).

### 2.5. Electrochemical Measurements Using GR Electrode Modified by PNC and Calculations

Experiments were performed by computerized potentiostat/galvanostat PGSTAT 302/Autolab with GPES 4.9 software, which was purchased from EcoChemie (Utrecht, The Netherlands), using cyclic voltammetry at scan rates (*v*) from 0.005 to 0.30 V s^−1^. All measurements were performed in three-electrode cells, with a working electrode—GR electrodes modified by PNC, a reference electrode—Ag/AgCl_(3 mol L_^−1^
_KCl)_ obtained from Metrohm (Herisau, Switzerland), and a counter electrode—platinum spiral. Electrochemical experiments were carried out in 0.05 mol L^−1^ SA buffer, pH 6.0, with 0.1 mol L^−1^ KCl and with or without 6 mmol L^−1^ PMS, which shifted the potential from −0.70 to 0.80 V vs. Ag/AgCl_(3 mol L_^−1^
_KCl)_ at room temperature.

Electrochemical detection of the analytical response was performed at a concentration interval from 0.10 to 48.4 mmol L^−1^ of glucose. To retain maximal electrode stability between electrochemical measurements, GR/PANI/AuNPs_(6nm)_-GOx, GR/Ppy/AuNPs_(6nm)_-GOx and GR/Ppy/AuNPs_(AuCl4_^−^_)_-GOx electrodes were stored at 4 °C in a closed vessel hanging over the solution of SA buffer, followed by a wash with distilled water before each measurement.

Cyclic voltammograms of all measurements were plotted and the parameters of Michaelis-Menten kinetics and the limit of detection (LOD) were estimated using statistic software SigmaPlot version 12.5. Calibration curves of all investigations were obtained while measuring analytical responses three times. The calculated LOD represents the lowest concentration of glucose, which provides an analytical signal greater than the background value plus 3 δ.

The principle schema of enzyme-assisted formation of PANI/AuNPs_(6nm)_-GOx, PANI/AuNPs_(AuCl4_^−^_)_-GOx, Ppy/AuNPs_(6nm)_-GOx and Ppy/AuNPs_(AuCl4_^−^_)_-GOx nanocomposites followed by electrochemical evaluation of the nanocomposite-modified electrodes are presented in Figure 1.

### 2.6. The Application of GR/Ppy/AuNPs_(AuCl4_^−^_)_-GOx Electrode for the Determination of Glucose in Human Serum

The human serum sample was diluted (1:10) in 0.05 mol L^−1^ SA, pH 6.0, and centrifuged at 14.6 × 10^3^ g. All investigations were performed using the GR electrode modified by Ppy/AuNPs_(AuCl4_^−^_)_-GOx nanocomposites. The electrochemical measurements were done in 10 times diluted human serum with 10 mmol L^−1^ of glucose before and after the addition of 1 mmol L^−1^ fructose, mannose, galactose, xylose, or saccharose. To evaluate the influence of ascorbic and uric acids on the developed biosensor, CV measurements were applied in 10 times diluted human serum with 10 mmol L^−1^ of glucose, 10 mmol L^−1^ of glucose and 0.01, 0.05 or 0.1 mmol L^−1^ of ascorbic acid (AA), and 10 mmol L^−1^ of glucose and 0.01 or 0.05 mmol L^−1^ of uric acid (UA).

## 3. Results and Discussion

### 3.1. Characteristics of Cyclic Voltammograms Registered Using GR/PANI/AuNPs_(6nm)_-GOx, GR/PANI/AuNPs_(AuCl4_^−^_)_-GOx, GR/Ppy/AuNPs_(6nm)_-GOx and GR/Ppy/AuNPs_(AuCl4_^−^_)_-GOx Electrodes

Novel nanocomposites based on AuNPs and polymers are able to improve various useful characteristics of biosensors [44]. The main aim of the present study was the investigation of a glucose biosensor based on enzyme-assisted-formed PANI/GOx and Ppy/GOx nanocomposites with incorporated gold nanoparticles. Enzyme-assisted polymerization/oligomerization depends on the concentration of polymerizable monomers in polymerization-bulk solution, the activity of the applied enzyme and the duration of the polymerization procedure [37,39].

Investigations of the shape of CV and the influence of scan rate on a shift of red-ox peaks provide some important information on the mechanism of electrochemical reaction and the rate of electron transfer from electrochemically active species [39]. The performance of the bare GR electrode and the same electrode, which was modified by PANI/AuNPs_(6nm)_-GOx, PANI/AuNPs_(AuCl4_^−^_)_-GOx, Ppy/AuNPs_(6nm)_-GOx and Ppy/AuNPs_(AuCl4_^−^_)_-GOx nanocomposites, was investigated by CV in 0.05 mol L^−1^ SA buffer, pH 6.0, in the presence of a soluble red-ox mediator—PMS. The reduction of the pyrrole monomer and the oxidation of chloroaurate ions proceeded simultaneously with the formation of Ppy and Au^0^ [43]. The influence of scan rate has been evaluated and is presented in Figure 2 and Table 1. No red-ox peaks were observed on the surface of the bare GR electrode when potential was swept in the range from −0.70 to 0.80 V. Potential diapason below 0.80 V was selected due the oxidation of gold at 1.10 V [5]. As is seen from Figure 2, well-defined anodic peaks were obtained on GR electrodes modified by PANI/AuNPs_(6nm)_-GOx, Ppy/AuNPs_(6nm)_-GOx and Ppy/AuNPs_(AuCl4_^−^_)_-GOx nanocomposites, instead of PANI/AuNPs_(AuCl4_^−^_)_-GOx-based nanocomposites. After the deposition of the nanocomposite’s layer on the working electrode, the form of registered cyclic voltammograms changed and became much wider than that before the modification. This effect illustrates a significant increase of the electrical capacitance of the modified electrode, and this fact is in line with previously published research on the evaluation of Ppy layers of different thickness and/or morphology [46].

It was determined that the diffusion barrier of GR electrodes, additionally modified by PANI or Ppy using enzyme-assisted polymerization, increased. It was observed that for GR/PANI/AuNPs_(6nm)_-GOx, GR/PANI/AuNPs_(AuCl4_^−^_)_-GOx, GR/Ppy/AuNPs_(6nm)_-GOx and GR/Ppy/AuNPs_(AuCl4_^−^_)_-GOx electrodes, *E*_pa_ has been shifted to more positive values, from −0.002 to 0.23 V, from 0.096 to 0.26 V, from −0.009 to 0.24 V and from −0.007 to 0.21 V respectively, when the scan rate increased from 0.005 to 0.30 V s^−1^. These red-ox peaks are attributed to the oxidation of glucose to gluconolactone, as it was reported previously [17], and indicates that the GR electrode modified by PNC with embedded GOx in the presence of AuNPs shows good electrocatalytic activity for electrochemical oxidation of glucose.

The electrochemical process is quasi-reversible and one cathodic peak has appeared on the GR electrode modified by PNC embedded with GOx and AuNPs on the reverse scan. It is seen that for GR/PANI/AuNPs_(6nm)_-GOx, GR/PANI/AuNPs_(AuCl4_^−^_)_-GOx, GR/Ppy/AuNPs_(6nm)_-GOx and GR/Ppy/AuNPs_(AuCl4_^−^_)_-GOx electrodes, *E*_pc_ has shifted towards more negative potentials, from −0.090 to −0.48 V, from −0.031 to −0.47 V, from −0.10 to −0.44 V and from −0.019 to −0.42 V respectively, by an increase of scan rate from 0.005 to 0.30 V s^−1^. The peak at potential of approximately −0.42 (−0.48) V for working electrodes may be observed due to the reduction of gold oxides, which occurs in potential of −0.40 V [17]. When the scan rate increased from 0.005 to 0.30 V s^−1^, anodic and cathodic peaks shifted for GR/PANI/AuNPs_(6nm)_-GOx by 0.23 V (Δ*E*_pa_ = *E*_pa(0.30 V s_^−1^_)_ − *E*_pa(0.005 V s_^−1^_)_) and by −0.39 V (Δ*E*_pc_ = *E*_pc(0.30 V s_^−1^_)_ − *E*_pc(0.005 V s_^−1^_)_) respectively, for GR/PANI/AuNPs_(AuCl4_^−^_)_-GOx—by 0.16 V and by −0.44 V, for GR/Ppy/AuNPs_(6nm)_-GOx—by 0.25 V and by −0.34 V, and for GR/Ppy/AuNPs_(AuCl4_^−^_)_-GOx—by 0.22 V and by −0.40 V. It is seen that in the case of GR/Ppy/AuNPs_(6nm)_-GOx and GR/Ppy/AuNPs_(AuCl4_^−^_)_-GOx, electrodes’ anodic peaks have shifted by 0.02 and 0.06 V towards more positive potential than that of GR/PANI/AuNPs_(6nm)_-GOx and GR/PANI/AuNPs_(AuCl4_^−^_)_-GOx electrodes, respectively. The form of anodic peaks of the GR/PANI/AuNPs_(AuCl4_^−^_)_-GOx electrode significantly differs from that of GR/PANI/AuNPs_(6nm)_-GOx, GR/Ppy/AuNPs_(6nm)_-GOx and GR/Ppy/AuNPs_(AuCl4_^−^_)_-GOx electrodes. Cathodic peaks registered using GR/PANI/AuNPs_(6nm)_-GOx and GR/PANI/AuNPs_(AuCl4_^−^_)_-GOx electrodes have shifted by 0.05 and 0.04 V towards more negative potential than that of GR/Ppy/AuNPs_(6nm)_-GOx and GR/Ppy/AuNPs_(AuCl4_^−^_)_-GOx electrodes (Table 1).

A diffusion-controlled, quasi-reversible electrochemical process was observed and the average value of the anodic and cathodic potential was calculated according to the methodology presented in Reference [10]. The formal potential (*E*^0^) at 0.30 V s^−1^ for GR/PANI/AuNPs_(6nm)_-GOx was −0.13 V, for GR/PANI/AuNPs_(AuCl4_^−^_)_-GOx was −0.11 V, for GR/Ppy/AuNPs_(6nm)_-GOx was −0.10 V and for GR/Ppy/AuNPs_(AuCl4_^−^_)_-GOx was −0.11 V. It indicates that the GR electrode modified by PNC nanocomposites containing GOx and AuNPs was characterized by a rather high degree of the reversibility and sufficient charge transfer rate. It was determined that the scan rate in the range from 0.10 to 0.30 V s^−1^ was the most optimal for accurate determination of red-ox peaks in cyclic voltammograms.

### 3.2. The Influence of Scan Rate on Red-ox Peaks of GR/PANI/AuNPs_(6nm)_-GOx, GR/PANI/AuNPs_(AuCl4_^−^_)_-GOx, GR/Ppy/AuNPs_(6nm)_-GOx and GR/Ppy/AuNPs_(AuCl4_^−^_)_-GOx Electrodes

Scan rate influences the electrochemical behavior of electrodes: a rather thick diffusion layer is formed when a slow scan rate is applied, and on the contrary, a thinner layer is formed when a fast scan rate is applied [47]. Anodic current (*I*_pa_) is proportional to the applied scan rate and the relationship between log *I*_pa_ and log *v* is linear when the “classic” electrochemical system is under investigation [39,47,48]. The influence of scan rate on the anodic current obtained on the surface of GR/PANI/AuNPs_(6nm)_-GOx, GR/PANI/AuNPs_(AuCl4_^−^_)_-GOx, GR/Ppy/AuNPs_(6nm)_-GOx and GR/Ppy/AuNPs_(AuCl4_^−^_)_-GOx electrodes in the absence and in the presence of glucose is presented in Figure 3A and Figure 4A, respectively. As is seen from the presented results, the value of anodic current for investigated systems has increased with the increasing scan rate.

When potential sweep rate increased from 0.005 to 0.30 V s^−1^ in the absence of glucose, the value of the anodic current obtained using GR/PANI/AuNPs_(6nm)_-GOx, GR/PANI/AuNPs_(AuCl4_^−^_)_-GOx, GR/Ppy/AuNPs_(6nm)_-GOx and GR/Ppy/AuNPs_(AuCl4_^−^_)_-GOx electrodes increased by 6.08 times (from 0.141 to 0.857 mA), 4.51 times (from 0.0732 to 0.330 mA), 8.63 times (from 0.0562 to 0.485 mA) and 2.92 times (from 0.0703 to 0.205 mA), respectively (Figure 3A and Figure 4A). However, in the presence of glucose, the value of *I*_pa_ registered using GR/PANI/AuNPs_(6nm)_-GOx, GR/PANI/AuNPs_(AuCl4_^−^_)_-GOx, GR/Ppy/AuNPs_(6nm)_-GOx and GR/Ppy/AuNPs_(AuCl4_^−^_)_-GOx electrodes increased by 7.41 times (from 0.147 to 1.09 mA), 4.75 times (from 0.0823 to 0.391 mA), 11.0 times (from 0.0788 to 0.870 mA) and 16.9 times (from 0.0343 to 0.578 mA) respectively, with the increase of scan rate from 0.005 to 0.30 V s^−1^ (Figure 3A and Figure 4A). The value of the anodic current for GR/PANI/AuNPs_(6nm)_-GOx and GR/PANI/AuNPs_(AuCl4_^−^_)_-GOx electrodes (Figure 3A) at the 0.30 V s^−1^ scan rate in the presence of glucose was 1.27 and 1.18 times higher in comparison with the anodic current registered for the same electrodes in the absence of glucose. Anodic currents for GR/Ppy/AuNPs_(6nm)_-GOx and GR/Ppy/AuNPs_(AuCl4_^−^_)_-GOx electrodes (Figure 4A) at the 0.30 V s^−1^ scan rate in the presence of glucose was 1.79 and 2.82 times higher than for the same electrodes without glucose. When the voltage sweep is faster, the red-ox reactions on the electrode do not have enough time to undergo completely [39]. Therefore, the scan rate of 0.10 V s^−1^ has been determined as the most optimal to achieve well-defined peaks in cyclic voltammograms and sufficiently high differences in red-ox peaks of cyclic voltammograms in the presence and absence of glucose.

The relationship between log *I*_pa_ and log *v* evaluated for GR/PANI/AuNPs_(6nm)_-GOx, GR/PANI/AuNPs_(AuCl4_^−^_)_-GOx, GR/Ppy/AuNPs_(6nm)_-GOx and GR/Ppy/AuNPs_(AuCl4_^−^_)_-GOx electrodes in the presence of glucose is presented in Figure 3B and Figure 4B, respectively. It is seen that in all cases, the logarithmic function of the anodic current was proportional to log *v* in the range of 0.699–2.48 mV s^−1^, which is in the agreement with data published in another paper [48]. The correlation coefficient of lines registered by GR/PANI/AuNPs_(6nm)_-GOx, GR/PANI/AuNPs_(AuCl4_^−^_)_-GOx, GR/Ppy/AuNPs_(6nm)_-GOx and GR/Ppy/AuNPs_(AuCl4_^−^_)_-GOx electrodes (Figure 4B) achieved 0.996, 0.997, 0.990 and 0.989, respectively. It allows us to conclude that the electrochemical reaction proceeded on GR/PANI/AuNPs_(6nm)_-GOx, GR/PANI/AuNPs_(AuCl4_^−^_)_-GOx, GR/Ppy/AuNPs_(6nm)_-GOx and GR/Ppy/AuNPs_(AuCl4_^−^_)_-GOx electrodes is diffusion-controlled and, therefore, such electrode can be applied for electrochemical biosensing of glucose in a rather broad concentration range. Mathematically, the slope of line (*a*) is determined by the equation of linear dependence (y = *a*x + *b*, where *a* = Δy/Δx = (y_2_ − y_1_)/(x_2_ − x_1_) = tan(*a*)). As is seen from Figure 3B, the slope of the line determined for the GR/PANI/AuNPs_(6nm)_-GOx electrode (tan(*a*) = 0.754) is 1.25 times steeper than that calculated for the GR/PANI/AuNPs_(AuCl4_^−^_)_-GOx electrode (tan(*a*) = 0.601). The same difference of the slope was determined for GR/Ppy/AuNPs_(6nm)_-GOx and GR/Ppy/AuNPs_(AuCl4_^−^_)_-GOx electrodes. As it is seen from Figure 4B, the slope of the line characterized for the GR/Ppy/AuNPs_(AuCl4_^−^_)_-GOx electrode (tan(*a*) = 0.781) is 1.12 times steeper in comparison with the results calculated for the GR/Ppy/AuNPs_(6nm)_-GOx electrode (tan(*a*) = 0.700). The steeper slope presents the increase of electron rate on the surface of GR/PANI/AuNPs_(6nm)_-GOx, GR/Ppy/AuNPs_(6nm)_-GOx and GR/Ppy/AuNPs_(AuCl4_^−^_)_-GOx electrodes. It is in agreement with a statement that AuNPs could enhance the sensitivity and stability of a glucose biosensor by catalyzing oxidation of H_2_O_2_, which is formed during enzymatic reaction, and effectively facilitate the electron transfer through a nanocomposite matrix due to good conductivity [2,4].

### 3.3. The Influence of the Duration of Enzymatic Polymerization on Cyclic Voltammogram Red-ox Peak Current Registered by GR/PANI/AuNPs_(6nm)_-GOx, GR/PANI/AuNPs_(AuCl4_^−^_)_-GOx, GR/Ppy/AuNPs_(6nm)_-GOx and GR/Ppy/AuNPs_(AuCl4_^−^_)_-GOx Electrodes

During enzyme-assisted polymerization, low molecular weight polymers remain in aqueous solutions while large molecular weight polymers are adsorbed on the electrode surface [49]. The efficiency of enzyme-assisted polymerization, the size of synthesized PNC and the conductivity strongly depend on the duration of polymerization [37]. Some researchers are reporting that with the increase of size from 39 to 1080 nm of Ppy nanotubes, which were decorated with AuNPs, the conductivity decreased from 75 to 0.24 S cm^−1^ [40]. However, the formation of larger PNC structures decreases the sensitivity of developed biosensors, due to the lower conductivity of Ppy nanotubes, which were decorated with gold particles [40]. In our previous research, we have reported that the optimal duration of enzyme-assisted polymerization of PNC was shorter than 4.5 days [44]. It was determined that by the increase of enzymatic synthesis duration from 1 to 4.5 days, the hydrodynamic diameter of PANI/AuNPs_(6nm)_-GOx, PANI/AuNPs_(AuCl4_^−^_)_-GOx, Ppy/AuNPs_(6nm)_-GOx and Ppy/AuNPs_(AuCl4_^−^_)_-GOx nanocomposites increased from 679 to 1128, from 570 to 659, from 512 to 594 and from 366 to 388 nm, respectively [50]. We predicted that the sensitivity of glucose biosensors developed in this paper based on GR/PANI/AuNPs_(6nm)_-GOx, GR/PANI/AuNPs_(AuCl4_^−^_)_-GOx, GR/Ppy/AuNPs_(6nm)_-GOx and GR/Ppy/AuNPs_(AuCl4_^−^_)_-GOx electrodes will be decreased when too-long polymerization duration is applied. Gold nanoparticles promote the electron transfer between glucose oxidase and the working electrode [27,51]. To evaluate the influence of enzyme-assisted polymerization on CV peak currents registered using GR/PANI/AuNPs_(6nm)_-GOx, GR/PANI/AuNPs_(AuCl4_^−^_)_-GOx, GR/Ppy/AuNPs_(6nm)_-GOx and GR/Ppy/AuNPs_(AuCl4_^−^_)_-GOx electrodes, the enzyme-assisted polymerization was performed for periods between 2 and 4 days in the dark at room temperature. To evaluate the anodic current of CV correctly, the difference of anodic peaks in the absence and the presence of 48 mmol L^−1^ glucose was calculated, and the obtained results are presented in Figure 5.

As it is seen from Figure 5, the value of the anodic current for GR/PANI/AuNPs_(6nm)_-GOx, GR/PANI/AuNPs_(AuCl4_^−^_)_-GOx, GR/Ppy/AuNPs_(6nm)_-GOx and GR/Ppy/AuNPs_(AuCl4_^−^_)_-GOx electrodes has decreased by 1.79 times (from 0.123 to 0.0687 mA), 1.72 times (from 0.0241 to 0.0140 mA), 7.69 times (from 0.233 to 0.0303 mA) and 11.7 times (from 0.263 to 0.0225 mA) respectively, with the increased duration of enzyme-assisted polymerization from 2 to 4 days. It led us to predict that a more sensitive glucose biosensor can be designed when enzyme-assisted polymerization lasting 2 days is applied for the formation of a nanocomposite-based layer.

The sensitivity of GR/PANI/AuNPs_(6nm)_-GOx, GR/PANI/AuNPs_(AuCl4_^−^_)_-GOx, GR/Ppy/AuNPs_(6nm)_-GOx and GR/Ppy/AuNPs_(AuCl4_^−^_)_-GOx electrodes calculated from data presented in Figure 5 was 0.0361, 0.0071, 0.0684 and 0.0772 mA mM cm^−2^, respectively. It was determined that after 2 days of enzyme-assisted polymerization, the sensitivity of the GR/PANI/AuNPs_(6nm)_-GOx electrode was 5.08 times higher, if compared with the GR/PANI/AuNPs_(AuCl4_^−^_)_-GOx electrode. Meanwhile, the sensitivity of the GR/Ppy/AuNPs_(6nm)_-GOx electrode was 1.13 times lower than that obtained on the GR/Ppy/AuNPs_(AuCl4_^−^_)_-GOx electrode. The values of the sensitivity for GR/PANI/AuNPs_(6nm)_-GOx and GR/PANI/AuNPs_(AuCl4_^−^_)_-GOx electrodes after 2 days of PNC formation were 1.89 and 10.9 times lower than that obtained using GR/Ppy/AuNPs_(6nm)_-GOx and GR/Ppy/AuNPs_(AuCl4_^−^_)_-GOx electrodes. To evaluate the role of AuNPs on the sensitivity in glucose sensing, the results obtained in the presence and absence of AuNPs in the structure of PANI and Ppy nanocomposites after 2 days of polymerization were compared. As can be seen from the results presented in Figure 5, the value of the anodic current for GR/PANI/GOx and GR/Ppy/GOx electrodes was 0.090 and 0.107 mA, respectively. The sensitivity of the GR/PANI/GOx (0.0264 mA mM cm^−2^) electrode was 1.37 times lower if compared with the GR/PANI/AuNPs_(6nm)_-GOx electrode. Meanwhile, the sensitivity of the GR/PANI/GOx electrode was 3.72 times higher in a comparison with the GR/PANI/AuNPs_(AuCl4_^−^_)_-GOx electrode, which were characterized as presented above. The sensitivity of the GR/Ppy/GOx (0.0314 mA mM cm^−2^) electrode was 2.18 and 2.46 times lower than that of GR/Ppy/AuNPs_(6nm)_-GOx and GR/Ppy/AuNPs_(AuCl4_^−^_)_-GOx electrodes, respectively. It indicates the advantage of polymeric nanocomposites with AuNPs. In any case, 2 days of enzyme-assisted polymerization is the most suitable duration for the development of GR/PANI/AuNPs_(6nm)_-GOx, GR/PANI/AuNPs_(AuCl4_^−^_)_-GOx, GR/Ppy/AuNPs_(6nm)_-GOx and GR/Ppy/AuNPs_(AuCl4_^−^_)_-GOx electrodes. The higher value of the anodic current calculated from differences of anodic peaks of CV in the presence and absence of glucose for GR/Ppy/AuNPs_(6nm)_-GOx and GR/Ppy/AuNPs_(AuCl4_^−^_)_-GOx electrodes is based on the conducting nature and increased ‘active surface area’ of nanocomposites, which offers more freedom regarding the orientation of entrapped GOx [10].

The surface concentration of the electrochemically active enzyme on the electrode surface (Γ) can be calculated by the following equation [10]:*I*_pa_ = n^2^F^2^*v*AΓ/4RT,(1)
where n is the number of electrons transferred (n = 2), *v* is the scan rate (0.1 V s^−1^) and A is the surface area of the working electrode (0.071 cm^2^), the constants R = 8.314 J K^−1^ mol^−1^, T = 294 K and F = 96485 C mol^−1^. The values of the surface coverage of GOx for GR/PANI/AuNPs_(6nm)_-GOx, GR/PANI/AuNPs_(AuCl4_^−^_)_-GOx, GR/Ppy/AuNPs_(6nm)_-GOx and GR/Ppy/AuNPs_(AuCl4_^−^_)_-GOx electrodes were evaluated as 4.54 × 10^−9^, 8.91 × 10^−10^, 8.63 × 10^−9^ and 9.71 × 10^−9^ mol cm^−2^ respectively, which are higher than that (1.76 × 10^−10^ mol cm^−2^) obtained on the glassy carbon electrode modified by GOx/P-_L_-Arg/f-MWCNTs composites [10]. Our results prove that the large surface area of Ppy/AuNPs_(6 nm)_ and Ppy/AuNPs_(AuCl4_^−^_)_ nanocomposites facilitate the high enzyme loading.

The morphology and size of PANI/AuNPs_(AuCl4_^−^_)_-GOx and Ppy/AuNPs_(AuCl4_^−^_)_-GOx nanocomposites obtained after 2 days of enzyme-assisted synthesis were evaluated by field emission scanning microscope and are presented in Figure 6. Poly-dispersed nanocomposites were obtained that tend to agglomerate.

As can be seen from Figure 6, PANI/AuNPs_(AuCl4_^−^_)_-GOx and Ppy/AuNPs_(AuCl4_^−^_)_-GOx nanocomposites had spherical structures. The size of PANI/AuNPs_(AuCl4_^−^_)_-GOx and Ppy/AuNPs_(AuCl4_^−^_)_-GOx nanocomposites was in the range of 300–452 nm and 80–113 nm respectively, and was similar to that of Ppy structures formed in the absence of AuNPs [50]. Huang et al. described the formation of 80–150 nm poly-dispersed spherical dendritic Au/Ppy nanocomposites by the use of self-assembly reaction during oxidation of pyrrole in the solution of HAuCl_4_ and toluene sulfonic acid [43]. The presented FE-SEM images show that the presence of AuCl_4_^−^ ions during the formation of polymeric nanocomposites increased the size of PANI/AuNPs_(AuCl4_^−^_)_-GOx and Ppy/AuNPs_(AuCl4_^−^_)_-GOx nanocomposites by 9 and 2 times if compared with the size of PANI/GOx and Ppy/GOx nanocomposites, which were synthesized in our previous research [52].

### 3.4. The Electrochemical Characterization of PANI/AuNPs_(6nm)_-GO_x_, PANI/AuNPs_(AuCl4_^−^_)_-GO_x_, Ppy/AuNPs_(6nm)_-GO_x_ and Ppy/AuNPs_(AuCl4_^−^_)_-GO_x_ Nanocomposites

The electrochemical characterization of PANI/AuNPs_(6nm)_-GO_x_, PANI/AuNPs_(AuCl4_^−^_)_-GO_x_, Ppy/AuNPs_(6nm)_-GO_x_ and Ppy/AuNPs_(AuCl4_^−^_)_-GO_x_ nanocomposites synthesized by the enzyme-assisted approach were investigated in 1.0 mol L^−1^ HCl by cyclic voltammetry performed in the potential range from −0.300 to 1.20 V. The cyclic voltammograms of bare GR electrodes and GR electrodes modified by PANI/AuNPs_(6nm)_-GO_x_, PANI/AuNPs_(AuCl4_^−^_)_-GO_x_, Ppy/AuNPs_(6nm)_-GO_x_ and Ppy/AuNPs_(AuCl4_^−^_)_-GO_x_ nanocomposites are presented in Figure 7.

As it is seen from Figure 7A,B (line 1), any red-ox peaks were observed on the surface of the bare GR electrode. However, in the case of the GR/PANI/AuNPs_(6nm)_-GOx electrode (Figure 7A, line 2), two oxidation peaks were registered at the 0.332 and 0.601 V values of potential, while for the GR/PANI/AuNPs_(6nm)_-GOx electrode (Figure 7A, line 3), only one oxidation peak at 0.349 V was observed. The conversion of PANI leucoemeraldine form to emeraldine salt showed the oxidation peaks at 0.332 and 0.349 V and it is in agreement with the results of other authors [53,54]. A 0.601 V potential peak in the case of the GR/PANI/AuNPs_(6nm)_-GOx electrode could be associated with the oxidation of emeraldine into pernigraniline form [53]. The CV obtained using GR/Ppy/AuNPs_(6nm)_-GO_x_ (Figure 7B, line 4) and GR/Ppy/AuNPs_(AuCl4_^−^_)_-GO_x_ (Figure 7B, line 5) electrodes showed the peaks at 0.425 and 0.545 V respectively, and may be associated with cation polaron’s transition into cation bipolaron’s state of polypyrrole [54].

The current density (mA cm^−2^) of GR/PANI/AuNPs_(6nm)_-GO_x_, GR/PANI/AuNPs_(AuCl4_^−^_)_-GO_x_, GR/Ppy/AuNPs_(6nm)_-GO_x_ and GR/Ppy/AuNPs_(AuCl4_^−^_)_-GO_x_ electrodes was calculated from CVs presented in Figure 7. The values of current of 0.227 and 0.410 mA at 0.332 and 0.349 V were determined for GR electrodes modified by PANI/AuNPs_(6nm)_-GO_x_ and PANI/AuNPs_(AuCl4_^−^_)_-GO_x_ (Figure 7A, lines 2 and 3). GR electrodes modified by Ppy/AuNPs_(6nm)_-GO_x_ and Ppy/AuNPs_(AuCl4_^−^_)_-GO_x_ were characterized by the current of 0.105 and 0.069 mA at 0.425 and 0.545 V (Figure 7B, lines 4 and 5). The current density of GR/PANI/AuNPs_(6nm)_-GO_x_, GR/PANI/AuNPs_(AuCl4_^−^_)_-GO_x_, GR/Ppy/AuNPs_(6nm)_-GO_x_ and GR/Ppy/AuNPs_(AuCl4_^−^_)_-GO_x_ electrodes was evaluated as 3.20, 5.78, 1.48 and 0.972 mA cm^−2^, and these values indicate rather high conductivity of formed structures, which increases the charge transfer ability of these PNC.

### 3.5. The Evaluation of the Stability and Analytical Characteristics of GR Electrodes Modified by PNC

The next stage of investigations was the evaluation of the stability of glucose biosensors based on GR electrodes modified by the developed PNC. For this purpose, PANI/AuNPs_(6nm)_-GOx, Ppy/AuNPs_(6nm)_-GOx and Ppy/AuNPs_(AuCl4_^−^_)_-GOx nanocomposites, which were formed by 2 days of enzyme-assisted polymerization, were evaluated. The application of PANI/AuNPs_(AuCl4_^−^_)_-GOx nanocomposites was refused due to the irregular form of cyclic voltammograms. The stability of the developed glucose biosensors was investigated within 6 or 22 days by the registration of CVs. GR electrodes modified by PANI/AuNPs_(6nm)_-GOx, Ppy/AuNPs_(6nm)_-GOx and Ppy/AuNPs_(AuCl4_^−^_)_-GOx nanocomposites (as described in Section 2.3) were hanged over 0.05 mol L^−1^ SA buffer, pH 6.0, solution at 4 °C between measurements. Registered anodic currents of prepared GR/PANI/AuNPs_(6nm)_-GOx, GR/Ppy/AuNPs_(6nm)_-GOx and GR/Ppy/AuNPs_(AuCl4_^−^_)_-GOx electrodes were equated to 100% and are presented in Figure 8.

The analytical response of GR/PANI/AuNPs_(6nm)_-GOx, GR/Ppy/AuNPs_(6nm)_-GOx and GR/Ppy/AuNPs_(AuCl4_^−^_)_-GOx electrodes towards glucose after storage for 6 days decreased down to 16.3%, 10.1% and 78.1%, respectively, of their initial value. It means that the value of the anodic current obtained on GR/PANI/AuNPs_(6nm)_-GOx, GR/Ppy/AuNPs_(6nm)_-GOx and GR/Ppy/AuNPs_(AuCl4_^−^_)_-GOx electrodes after 6 days was 6.13, 9.90 and 1.28 times lower than that immediately after the design of electrodes. It is seen that electrochemical biosensors based on the GR/Ppy/AuNPs_(AuCl4_^−^_)_-GOx electrode were 4.79 and 7.73 times more stable than those based on GR/PANI/AuNPs_(6nm)_-GOx and GR/Ppy/AuNPs_(6nm)_-GOx electrodes. The further investigation was performed with 2 days of enzyme-assisted synthesized Ppy/AuNPs_(AuCl4_^−^_)_-GOx nanocomposites due to their high stability and the simplicity of surface modification of the GR electrode. The *τ*_1/2_ (50% of initial response) for the developed GR/Ppy/AuNPs_(AuCl4_^−^_)_-GOx electrode was 19 days.

Ppy/AuNPs_(AuCl4_^−^_)_-GOx nanocomposites formed on the surface of the GR electrode improved the efficiency of charge transfer from immobilized GOx and maintained their bio-catalytic activity. During electrochemical measurements, electrons are transferred toward the positively charged surface of the GR electrode and current is registered. In the presence of glucose and dissolved oxygen in the solution of SA buffer, immobilized on the GR electrode, GOx contained in polymeric nanocomposites generates hydrogen peroxide and gluconolactone, which is hydrolyzed to gluconic acid (Figure 1). PMS is able to re-oxidase the red-ox active center of GOx and the electrons are transferred via a reduced form of the mediator PMSH_2_, in two ways: (i) directly to the GR electrode and (ii) through AuNPs [26]. AuNPs are able to facilitate an electron transfer between protein, the red-ox mediator and the GR electrode, and to improve the sensitivity of the developed analytical system [26,51]. To evaluate the analytical characteristics of the developed electrochemical biosensor based on the GR/Ppy/AuNPs_(AuCl4_^−^_)_-GOx electrode, anodic current responses were measured by CV at the glucose concentration range from 0.10 to 48.4 mmol L^−1^. The anodic current of the GR/Ppy/AuNPs_(AuCl4_^−^_)_-GOx electrode increased due to the GOx-catalyzed reaction and the increased concentration of oxidizable products (Figure 9A). As it is seen from the presented CVs (Figure 9A), the GR/Ppy/AuNPs_(AuCl4_^−^_)_-GOx electrode is also involved in the electrochemical reduction of O_2_ and H_2_O_2_ and it is in line with findings presented by some other researchers, which evaluated graphene/nano-Au/GOx systems immobilized on the GC electrode [19]. The anodic peak registered for the GR/Ppy/AuNPs_(AuCl4_^−^_)_-GOx electrode appeared at 0.138 V, while the cathodic peak appeared at −0.343 V. 

The GR/Ppy/AuNPs_(AuCl4_^−^_)_-GOx electrode can be used for glucose determination due to the increase of the anodic current of CVs (Figure 9A). The amount of glucose being oxidized by GOx immobilized on the electrode is proportional to the concentration of glucose present in sample solution [18,19]. Hyperbolic dependence, which is presented in Figure 9B, is in agreement with Michaelis-Menten kinetics. The maximal current generated during the enzyme-catalyzed reaction (*I*_max_) and the apparent Michaelis constant (*K*_M(apparent)_) were correspondingly *a* and *b* parameters of hyperbolic function *y* = *ax*/(*b* + *x*), which has been used for the approximation of results. The reported glucose biosensor based on the GR/Ppy/AuNPs_(AuCl4_^−^_)_-GOx electrode was characterized by Δ*I*_max_ = 0.292 mA and *K*_M(apparent)_ = 0.348 mmol L^−1^. A low value of *K*_M(apparent)_ indicates high affinity of glucose towards immobilized glucose oxidase [27].

Analytical measurements of the GR/Ppy/AuNPs_(AuCl4_^−^_)_-GOx electrode display that the linear detection range for the determination of glucose was up to 0.70 mmol L^−1^, with the correlation coefficient of 0.9895 (Figure 9C). The sensitivity of the developed glucose biosensor in the presence of 0.70 mmol L^−1^ glucose was determined as 4.31 mA mM cm^−2^. The sensitivity of the GR/Ppy/AuNPs_(AuCl4_^−^_)_-GOx electrode at a low concentration of glucose was higher than that at a high concentration. Lower sensitivity (0.0772 mA mM cm^−2^) at high concentrations (48.4 mmol L^−1^) of glucose than that at low concentrations could be related to two factors: (i) the slower diffusion of glucose, because at higher glucose concentrations the viscosity of glucose solution increases, and (ii) adsorption/desorption dynamics of glucose at high concentrations [19]. The evaluated sensitivity was higher than that determined by other researchers on the surfaces of the GC electrode modified by GOx/P-L-Arg/f-MWCNTs composites (48.86 μA mM cm^−2^) [10] and of the GC electrode modified by GOx-poly(L-lysine)/reduced graphene oxide-ZrO_2_ composites (11.65 μA mM cm^−2^) [15]. The relative standard deviation of the analytical signal at glucose concentration of 0.70 mmol L^−1^ on the GR/Ppy/AuNPs_(AuCl4_^−^_)_-GOx electrode was 13.9%. The value of the LOD for the developed glucose biosensor was determined as 0.10 mmol L^−1^. The calculated LOD is in the same range as that determined by some other researchers for the GC electrode modified by GOx/P-_L_-Arg/f-MWCNTs composites (0.10 mmol L^−1^) [10], and lower than that calculated for the GC electrode modified by GOx-poly(L-lysine)/reduced graphene oxide-ZrO_2_ composites (0.13 mmol L^−1^) [15]. According to the statement by Vilian et al. [15], the increase of the electron transfer kinetics for glucose biosensing and electrocatalytic activity can be determined by high surface area of gold nanostructures embedded within polymers.

The high sensitivity and the low limit of detection are significant advantages of the GR electrode modified by Ppy/AuNPs_(AuCl4_^−^_)_-GOx nanocomposites, even if the linear range of the developed glucose biosensor was lower than the physiological level of glucose in biological samples, in the range from 3 to 8 mmol L^−1^ [55,56] (in the blood serum for healthy persons: 3.9–5.6 mmol L^−1^).

### 3.6. The Application of the Developed Glucose Biosensor Based on GR Electrode Modified by Ppy/AuNPs_(AuCl4_^−^_)_-GOx Nanocomposites for Glucose Determination in the Sample of Human Serum

The addition of the interfering species, such as 1.0 mmol L^−1^ of fructose, mannose, xylose or saccharose, in the solution of 10.0 mmol L^−1^ glucose do not show the influence on the registered current of glucose (Figure 10A), while the addition of 1.0 mmol L^−1^ galactose in the solution of 10.0 mmol L^−1^ glucose decreased the registered current of glucose by 5.81% in comparison with results obtained for the solution, which contained only glucose without galactose. The influence of ascorbic and uric acids on the determination of glucose is presented in Figure 10B. It is seen that AA and UA have negligible effects on the registered current: after the addition of 10.0 mmol L^−1^ glucose solution with 0.01, 0.05 or 0.1 mmol L^−1^ of AA, registered responses changed by 3.23%, 4.40% and 6.16%, while after the addition of 10.0 mmol L^−1^ glucose solution with 0.01 or 0.05 mmol L^−1^ of UA, by 2.19% or 13.4%, if compared with the current registered to glucose without electroactive species. The investigated concentrations of AA and UA were higher than normal physiological concentrations (0.141 mmol L^−1^ of AA [57] and 0.1 mmol L^−1^ of UA [55]) in human serum after 10 times dilution. These statements indicate the suitability of the developed glucose biosensor based on the GR electrode modified by Ppy/AuNPs_(AuCl4_^−^_)_-GOx nanocomposites for practical application.

The suitability of the developed biosensor based on the GR electrode modified by Ppy/AuNPs_(AuCl4_^−^_)_-GOx nanocomposites for glucose determination in the samples of human serum was performed. For this purpose, human serum 10 times diluted with 0.05 mol L^−1^ SA buffer solution, pH 6.0, with known initial glucose concentrations, was investigated. Then, various amounts of glucose were added in order to imitate samples with increasing glucose concentrations in the linear detection range of the analyte. Each sample of human blood serum was assessed at least three or four times, and the achieved results were expressed by the average values and are presented in Table 2. It was found that recoveries of developed glucose biosensor in the sample of human serum were in the range of 93.6–94.8%.

The response of CV was registered in human blood serum 10 times diluted with 0.05 mol L^−1^ SA buffer, pH 6.0, sample containing 6 mmol L^−1^ PMS.

The advantages of the developed biosensor based on the graphite rod electrode modified by Ppy/AuNPs_(AuCl4_^−^_)_-GO_x_ nanocomposites are: (i) the high sensitivity (4.31 mA mM cm^−2^) and rather good stability (19 days), (ii) the low limit of detection (0.10 mmol L^−1^), (iii) the applicability in multiple analyses, (iv) the low price of a single analysis, (v) the short duration of a single measurement (20 s) and (vi) the high resistance to interfering materials, which allows to apply modified GR electrodes for glucose biosensing in clinical practice. The development of a glucose biosensor without a red-ox mediator is expected to be the next step of our investigations in this research area.

## 4. Conclusions

PANI and Ppy nanocomposites with GOx and AuNPs were formed by enzyme-assisted synthesis of PANI and Ppy and were deposited on the surface of GR electrodes. The relationship between log *I*_pa_ and log *v* registered from CV anodic peaks in the presence of glucose was linear and the electrochemical reaction was controlled by the diffusion. The glucose biosensor based on GR modified by Ppy/AuNPs_(AuCl4_^−^_)_-GOx nanocomposites was characterized by high sensitivity (4.31 mA mM cm^−2^) and good stability (the half-life period of 19 days). The developed glucose biosensor based on the GR electrode modified by Ppy/AuNPs_(AuCl4_^−^_)_-GOx nanocomposites was suitable for the determination of glucose in the sample of human blood serum in the presence of interfering species with the recoveries in the range of 93.6–94.8%. The presented bioelectrochemical system can be useful for the development of biosensors, biofuel cells and bioelectronics devices.

## Figures and Tables

**Figure 1 polymers-12-03026-f001:**
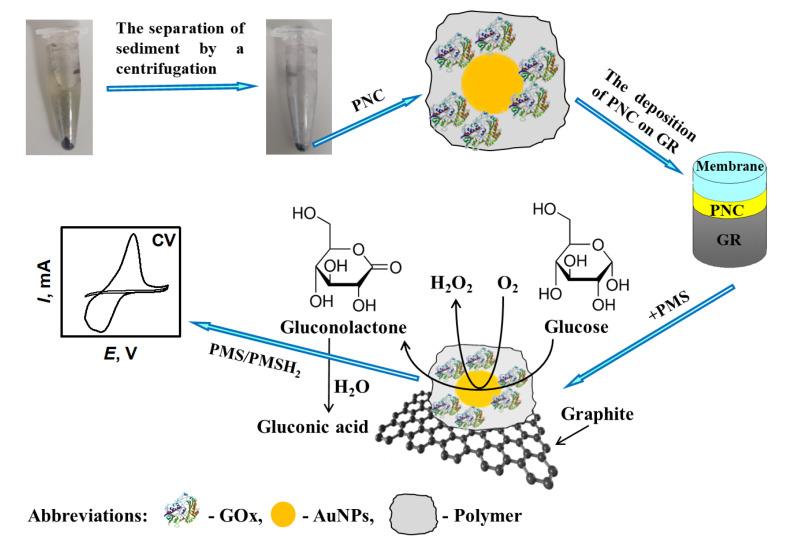
The principle scheme of PANI/AuNPs_(6nm)_-GOx, PANI/AuNPs_(AuCl4_^−^_)_-GOx, Ppy/AuNPs_(6nm)_-GOx and Ppy/AuNPs_(AuCl4_^−^_)_-GOx nanocomposites’ formation, followed by investigations using cyclic voltammetry.

**Figure 2 polymers-12-03026-f002:**
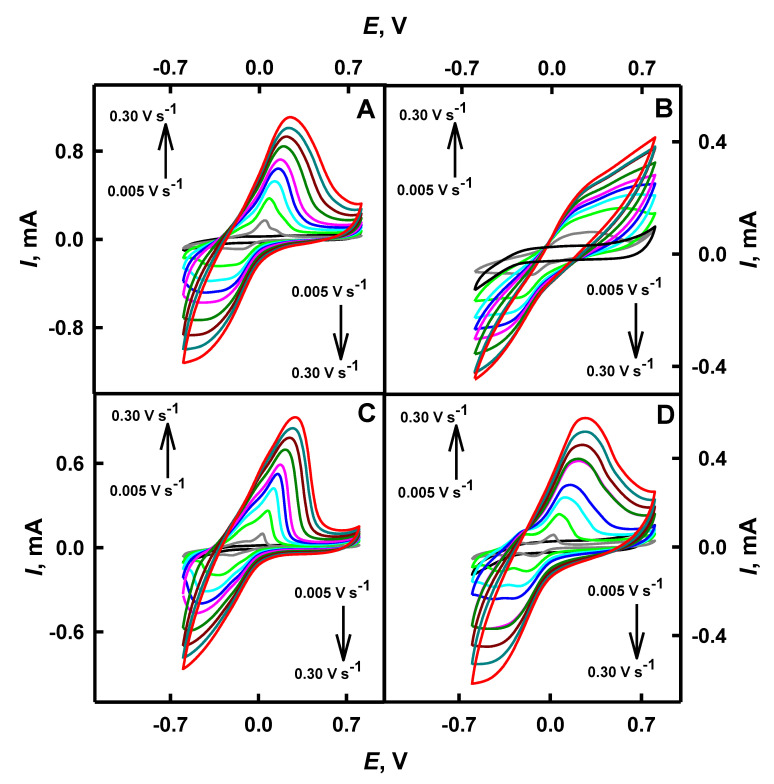
Cyclic voltammograms registered by GR electrodes modified with: PANI/AuNPs_(6nm)_-GOx (**A**), PANI/AuNPs_(AuCl4_^−^_)_-GOx (**B**), Ppy/AuNPs_(6nm)_-GOx (**C**) and Ppy/AuNPs_(AuCl4_^−^_)_-GOx (**D**) nanocomposites at a potential scan rate from 0.005 to 0.03 V s^−1^. Conditions: analytical signal was registered in 0.05 mol L^−1^ SA buffer, pH 6.0, with 0.01 mol L^−1^ KCl, without (black colored line) and with (all other colored lines) 6 mmol L^−1^ of PMS and 48 mmol L^−1^ of glucose.

**Figure 3 polymers-12-03026-f003:**
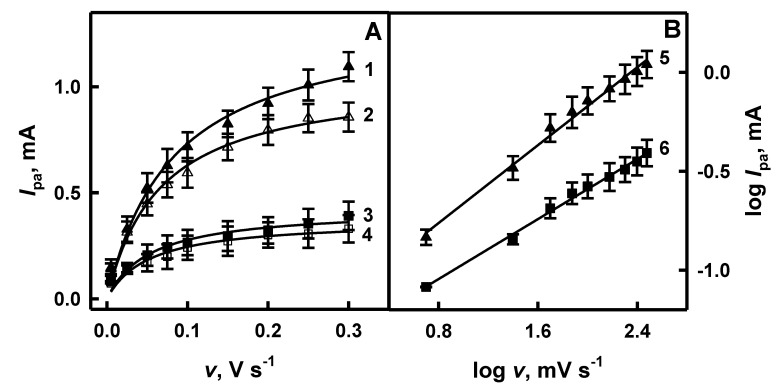
The influence of scan rate on the registered anodic current (**A**) and the relationship between log *I*_pa_ and log *v* (**B**) for the GR/PANI/AuNPs_(6nm)_-GOx and GR/PANI/AuNPs_(AuCl4_^−^_)_-GOx electrodes in the absence (hollow symbols, curves 2 and 4) and in the presence (filled symbols, curves 1 and 3) of glucose. Curves 1, 2 and 5 were registered using the GR/PANI/AuNPs_(6nm)_-GOx electrode, and curves 3, 4 and 6 using the GR/PANI/AuNPs_(AuCl4_^−^_)_-GOx electrode. Analytical signals (differences of anodic peak currents) were registered in 0.05 mol L^−1^ SA buffer, pH 6.0, containing 0.01 mol L^−1^ of KCl and 6 mmol L^−1^ of PMS without glucose and in the presence of 48 mmol L^−1^ glucose.

**Figure 4 polymers-12-03026-f004:**
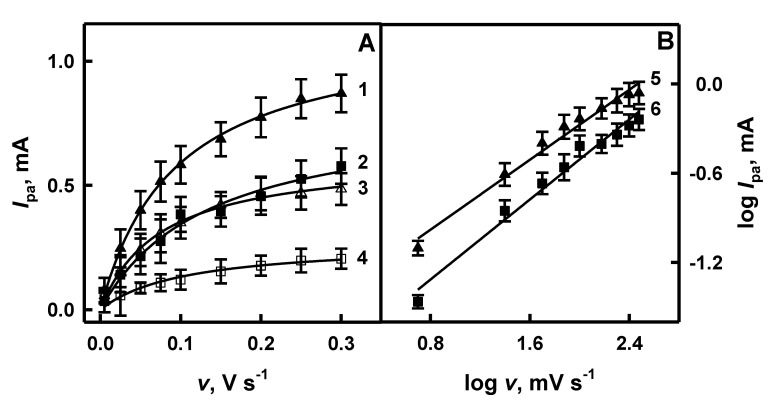
The influence of scan rate on the registered anodic current (**A**) and the relationship between log *I*_pa_ and log *v* (**B**) for the GR/Ppy/AuNPs_(6nm)_-GOx and GR/Ppy/AuNPs_(AuCl4_^−^_)_-GOx electrodes in the absence (hollow symbols, curves 3 and 4) and in the presence (filled symbols, curves 1 and 2) of glucose. Curves 1, 3 and 5 were registered using the GR/Ppy/AuNPs_(6nm)_-GOx electrode, and curves 2, 4 and 6 using the GR/Ppy/AuNPs_(AuCl4_^−^_)_-GOx electrode. Analytical signals (differences of anodic peak currents) were registered in 0.05 mol L^−1^ SA buffer, pH 6.0, containing 0.01 mol L^−1^ of KCl and 6 mmol L^−1^ of PMS without glucose and in the presence of 48 mmol L^−1^ glucose.

**Figure 5 polymers-12-03026-f005:**
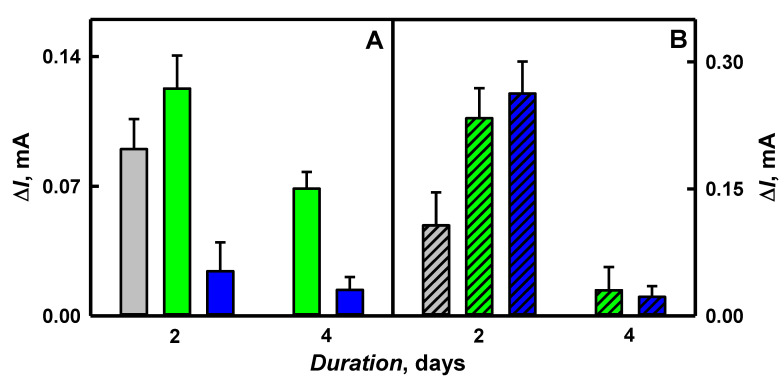
The diagrams of CV anodic peak currents registered for GR/PANI-GOx, GR/PANI/AuNPs_(6nm)_-GOx, GR/PANI/AuNPs_(AuCl4_^−^_)_-GOx (**A**) and for GR/Ppy-GOx, GR/Ppy/AuNPs_(6nm)_-GOx, GR/Ppy/AuNPs_(AuCl4_^−^_)_-GOx (**B**) electrodes covered by a nanocomposite layer after 2 and 4 days of enzyme-assisted polymerization. Conditions: grey (**A**) and grey dashed (**B**) columns are presented for GR/PANI-GOx and GR/Ppy-GOx electrodes, green (**A**) and green dashed (**B**) columns for GR/PANI/AuNPs_(6nm)_-GOx and GR/Ppy/AuNPs_(6nm)_-GOx electrodes, blue (**A**) and blue dashed (**B**) columns for GR/PANI/AuNPs_(AuCl4_^−^_)_-GOx (**A**) and GR/Ppy/AuNPs_(AuCl4_^−^_)_-GOx (**B**) electrodes. Analytical signals (differences of anodic peak currents) were registered in 0.05 mol L^−1^ SA buffer, pH 6.0, containing 0.01 mol L^−1^ of KCl, 6 mmol L^−1^ of PMS and 48 mmol L^−1^ of glucose.

**Figure 6 polymers-12-03026-f006:**
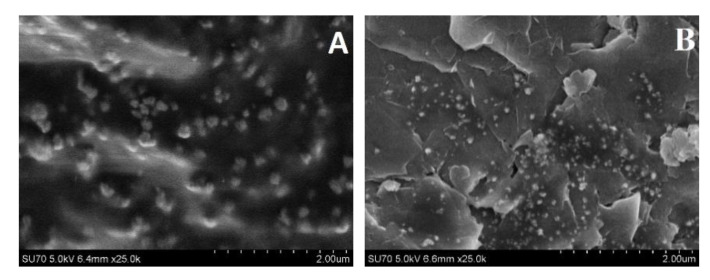
Field emission scanning electron microscope (FE-SEM) images of the PANI/AuNPs_(AuCl4_^−^_)_-GOx composite (**A**) and the Ppy/AuNPs_(AuCl4_^−^_)_-GOx composite (**B**) formed after 2 days of enzyme-assisted formation.

**Figure 7 polymers-12-03026-f007:**
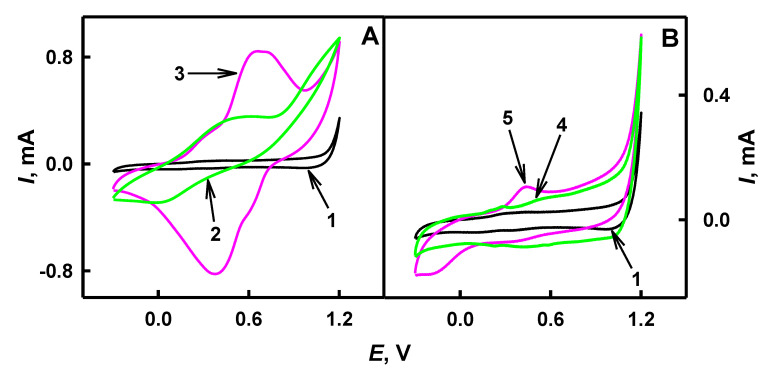
Cyclic voltammograms of GR modified by PANI (**A**) and Ppy (**B**) nanocomposites with GOx and AuNPs. Conditions: black lines—bare GR (**A**,**B**, 1) electrode, green lines—GR/PANI/AuNPs_(6nm)_-GOx (**A**, 2) and GR/Ppy/AuNPs_(6nm)_-GOx (**B**, 4) electrodes, pink lines—GR/PANI/AuNPs_(AuCl4_^−^_)_-GOx (**A**, 3) and GR/Ppy/AuNPs_(AuCl4_^−^_)_-GOx (**B**, 5) electrodes. The cyclic voltammograms were registered in 1.0 mol L^−1^ HCl at a potential sweep rate of 0.10 V s^−1^.

**Figure 8 polymers-12-03026-f008:**
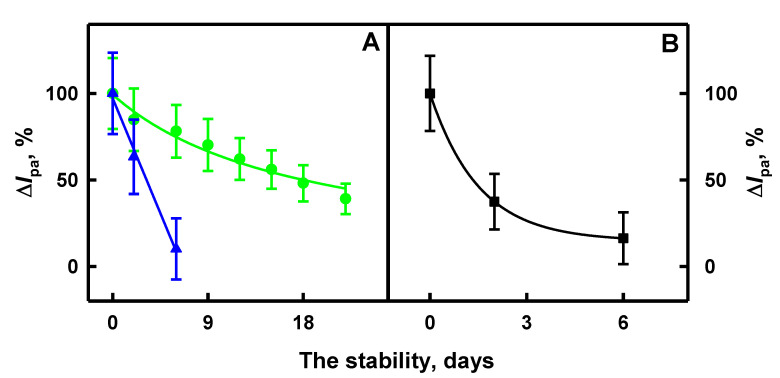
The stability of GR electrodes modified by Ppy/AuNPs_(AuCl4_^−^_)_-GOx ((**A**), green colored curve), Ppy/AuNPs_(6nm)_-GOx ((**A**), blue colored curve) and PANI/AuNPs_(6nm)_-GOx ((**B**), black colored curve) nanocomposites formed by 2 days of enzyme-assisted polymerization. Analytical signals, which were based on the comparison of oxidation peaks of CVs, were registered in 0.05 mol L^−1^ SA buffer, pH 6.0, containing 0.01 mol L^−1^ KCl in the presence of 6 mmol L^−1^ PMS and 9.92 mmol L^−1^ of glucose.

**Figure 9 polymers-12-03026-f009:**
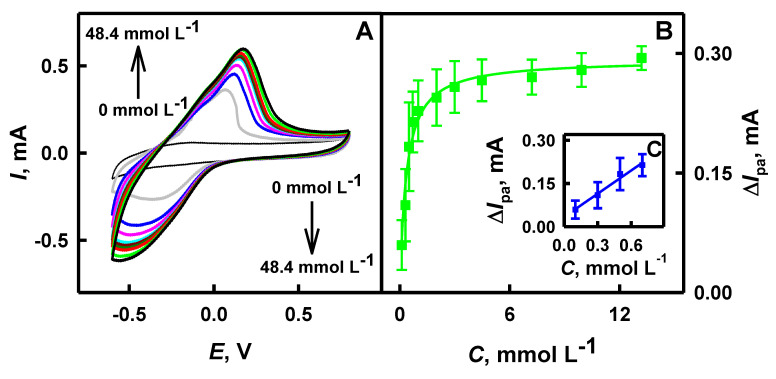
Cyclic voltammograms (**A**), calibration plot (**B**) and the linear detection range (**C**) of the GR/Ppy/AuNPs_(AuCl4_^−^_)_-GOx electrode in the presence of glucose. Conditions: (**A**) CVs without (thin black cyclic voltammogram) and with (thick colored cyclic voltammograms) 6 mmol L^−1^ of PMS, (**B**, green colored curve) the dependence of the analytical signal on glucose concentration in the range from 0.10 to 13.2 mmol L^−1^ of glucose, (**C**, blue colored curve) dependence of the analytical signal on glucose concentration in the range from 0.10 to 0.7 mmol L^−1^ of glucose, for 2 days of enzyme-assisted polymerization. Analytical signals, which were based on the comparison of oxidation peaks of CVs in the presence and absence of glucose, were registered in 0.05 mol L^−1^ SA buffer, pH 6.0, containing 0.01 mol L^−1^ KCl in the presence of corresponding glucose concentration.

**Figure 10 polymers-12-03026-f010:**
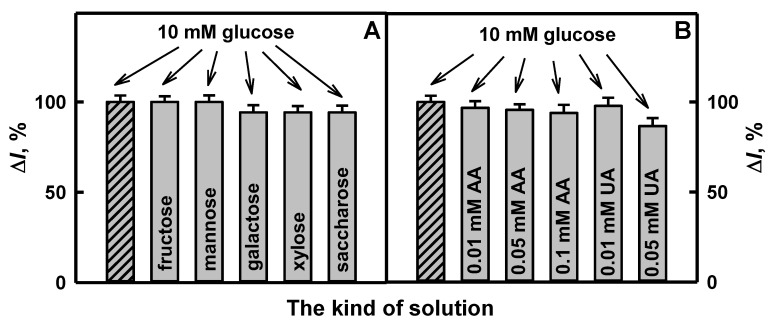
The effect of interfering substances on the CV response of the developed glucose biosensor based on the GR electrode modified by Ppy/AuNPs_(AuCl4_^−^_)_-GOx nanocomposites. Diagrams of registered current in 10 times diluted human serum sample after the addition of 10.0 mmol L^−1^ glucose (dashed column) with (**A**) 1.0 mmol L^−1^ of fructose, mannose, galactose, xylose or saccharose, and (**B**) with 0.01, 0.05 or 0.1 mmol L^−1^ of AA and 0.01 or 0.05 mmol L^−1^ of UA. The response of CV was registered in human blood serum 10 times diluted with 0.05 mol L^−1^ SA buffer, pH 6.0, containing 6.0 mmol L^−1^ of PMS.

**Table 1 polymers-12-03026-t001:** The electrochemical characteristics of red-ox peaks obtained on GR electrodes modified by PANI/AuNPs_(6nm)_-GOx, PANI/AuNPs_(AuCl4_^−^_)_-GOx, Ppy/AuNPs_(6nm)_-GOx and Ppy/AuNPs_(AuCl4_^−^_)_-GOx nanocomposites at applied CV scan rates in the range from 0.005 to 0.30 V s^−1^. Conditions are the same as in Figure 2.

The Kind of Polymer Nanocomposites on GR	*E*_pa_, V	*E*_pc_, V	Δ*E*_pa_, V	Δ*E*_pc_, V
PANI/AuNPs_(6nm)_-GOx	−0.002–(0.23)	−0.090–(−0.48)	0.23	−0.39
PANI/AuNPs_(AuCl4_^−^_)_-GOx	+0.096–(0.26)	−0.031–(−0.47)	0.16	−0.44
Ppy/AuNPs_(6nm)_-GOx	−0.009–(0.24)	−0.10–(−0.44)	0.25	−0.34
Ppy/AuNPs_(AuCl4_^−^_)_-GOx	−0.007–(0.21)	−0.019–(−0.42)	0.22	−0.40

**Table 2 polymers-12-03026-t002:** Recovery of glucose in the sample of human serum using a biosensor based on the GR electrode modified by Ppy/AuNPs_(AuCl4_^−^_)_-GOx nanocomposites.

Added Concentration, mmol L^−1^	Detected Concentration, mmol L^−1^	Recovery, %	Number of Measurements
0.25	0.234	93.6	4
0.30	0.280	93.3	3
0.33	0.306	92.7	3
0.46	0.436	94.8	3

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
