# Peer review of "Formation and Electrochemical Evaluation of Polyaniline and Polypyrrole Nanocomposites Based on Glucose Oxidase and Gold Nanostructures"

_polymers, 2020, doi:10.3390/polym12123026_

Round 1

Reviewer 1 Report

In this work, the authors reported the synthesis of nanocomposites based on two conducting polymers (polyaniline (PANI) and polypyrrole (Ppy)) with embedded glucose oxidase (GOx) and 6 nm size gold nanoparticles (AuNPs(6nm)) or gold-nanoclusters formed from chloroaurate ions (AuCl4-) by enzyme-assisted  polymerization. The combination of mechanistic model with cyclic voltammetry (CV) as analysis method was performed and it was applied for the evaluation of analytical characteristics of glucose biosensor.

I recommend enhancement of result and manuscript before publication, and the following point needs to be explained properly:

  1. Morphology characterization for the synthesized PNC should be added.

  1. What would be the results if the size gold nanoparticles was 14 nm as reported in their previous work?

  1. The detection of glucose in the real serum samples is suggested.

This paper needs major revision before it is suitable for publication in Polymers.

Author Response

See attached response to reviewer 1.

Reviewer 2 Report

Polyaniline and polypyrrole nanocomposites with embedded glucose oxidase in the presence of gold nanoparticles were enzymatically formed and investigated on the surface of graphite rod electrodes

For an easier understanding of the processes, I strongly recommend the authors to prepare a schema of the working steps - as graphical abstracts, maybe....

The introduction needs to be refined a little bit in order to prove more comparative data in an easier form to compare - I recommend a table underlining the parameters/conditions/sensitivities...

Figure 5 - too high, maybe coloured bars are more helpful than the models chosen by the authors

Figure 6 - too high - the "graph on the graph" chosen by the authors needs to  be clarified... hard to understand the meaning.. also here I think that coloured labels could be more helpful for a better understanding

Figure 7- see the previous comment for the second graph 7B...

Author Response

See attached response to reviewer 2.

Reviewer 3 Report

German et. al., submitted a paper entitled “Formation and electrochemical evaluation of polypyrrole nanocomposites based on glucose oxidase and gold-nanostructures” to “Polymers (I.F= 3.426)”. In this manuscript they have demonstrated the electrochemical sensing abilities of polymers containing composite electrodes. The work is impressive, but still need to undergo major revision.

  1. Author performed all the electrochemical investigations in detail and provided enough data. But, regarding the characterization of electrodes, there is no information provided, which is essential to clarify their results.
  2. SEM or AFM images of the electrodes ((PANI/AuNPs(6nm)-GOx, PANI/AuNPs(AuCl4-)-GOx, Ppy/AuNPs(6nm)-GOx and Ppy/AuNPs(AuCl4-)-GOx) are essential to understand the morphological changes as well as the presence of AuNPs.
  3. There is no comparative experiment done without AuNPs or without those polymers, which may verify the role of polymers or AuNPs in glucose sensing.
  4. What happened to the sensitivity in presence of competitor like Ascorbic acid, Fructose etc.
  5. The mechanism behind the exceptional response of GR modified Ppy/AuNPs(AuCl4-)-GOx is still need more clarification with evidence.
  6. Author must concise the conclusion part with only essential finding information.
  7. Since many electrochemical glucose sensors were already available, author must state the merits of their work in the conclusion part.

Author Response

See attached response to reviewer 3.

Round 2

Reviewer 1 Report

The authors had carefully revised the manuscript. My concerns and questions had been addressed. Its publication in Polymers is recommended.

Author Response

Response to reviewer #1:

We would like to thank the reviewer for very professional review of our manuscript, valuable comments and recommendations. Thank you for pointing out our mistakes and giving suggestions which further on improve clarity of this paper.

Comment by Reviewer 1:

The authors had carefully revised the manuscript. My concerns and questions had been addressed. Its publication in Polymers is recommended.

Response from authors:

Thank you for your careful review and favourable feedback.

We will thank for positive feedback and valuable recommendations.

We hope after all these corrections our manuscript is suitable for publication.

Yours sincerely,
Arunas Ramanavicius

----------------------------------------------------------------
Prof. habil. dr. Arunas Ramanavicius

Head of Department of Physical Chemistry,

Faculty of Chemistry, Vilnius University,

Naugarduko 24, 03225 Vilnius 6, Lithuania; e-mail: [email protected]

Reviewer 3 Report

Author Revised the manuscript according to the comments, this work can be accepted after addressing the following queries.

  1. Author must provide error bars for Figures 3, 4 and 10.
  2. The SEM images (Figure. 6) must be in same scale bars or otherwise mention the scale bars in the Figure caption.

Author Response

Dear reviewer, please see attached file with attached figures.

Response to reviewer #3:

We would like to thank the reviewer for very professional review of our manuscript, valuable comments and recommendations. Thank you for pointing out our mistakes and giving suggestions which further on improve clarity of this paper. We did our best in order to improve the manuscript according to comments and recommendations. All the most important changes are highlighted in the revised manuscript. Corrections and changes are highlighted in the manuscript (in red).

Please find below short explanations and answers to your questions:

Comment by Reviewer 3:

  1. Author must provide error bars for Figures 3, 4 and 10.

Response from authors:

Error bars were provided for Figures 3, 4 and10 and added to this paper.

Figure 3. The influence of scan rate on the registered anodic current (A) and the relationship between log Ipa and log v (B) for the GR/PANI/AuNPs(6nm)-GOx and GR/PANI/AuNPs(AuCl4-)-GOx electrodes in the absence (hollow symbols, curves 2 and 4) and in the presence (filled symbols, curves 1 and 3) of glucose. Curves 1,2 and 5 were registered using GR/PANI/AuNPs(6nm)-GOx electrode, and curves 3,4 and 6 – using GR/PANI/AuNPs(AuCl4-)-GOx electrode. Analytical signals (differences of anodic peak currents) were registered in 0.05 mol L-1 SA buffer, pH 6.0, containing 0.01 mol L-1 of KCl and 6 mmol L-1 of PMS without glucose and in the presence of 48 mmol L-1 glucose.

Figure 4. The influence of scan rate on the registered anodic current (A) and the relationship between log Ipa and log v (B) for the GR/Ppy/AuNPs(6nm)-GOx and GR/Ppy/AuNPs(AuCl4-)-GOx electrodes in the absence (hollow symbols, curves 3 and 4) and in the presence (filled symbols, curves 1 and 2) of glucose. Curves 1,3 and 5 were registered using GR/Ppy/AuNPs(6nm)-GOx electrode, and curves 2,4 and 6 – using GR/Ppy/AuNPs(AuCl4-)-GOx electrode. Analytical signals (differences of anodic peak currents) were registered in 0.05 mol L-1 SA buffer, pH 6.0, containing 0.01 mol L-1 of KCl and 6 mmol L-1 of PMS without glucose and in the presence of 48 mmol L-1 glucose.

Figure 10. The effect of interfering substances on the CV response of the developed glucose biosensor based on GR electrode modified by Ppy/AuNPs(AuCl4-)-GOx nanocomposites. Diagrams of registered current in 10 times diluted human serum sample after the addition of 10.0 mmol L-1 glucose (dashed column) with (A) 1.0 mmol L-1 of fructose, mannose, galactose, xylose or saccharose; (B) – 0.01, 0.05 or 0.1 mmol L-1 of AA and 0.01 or 0.05 mmol L-1 of UA. The response of CV was registered in human blood serum 10 times diluted with 0.05 mol L-1 SA buffer, pH 6.0, containing 6.0 mmol L-1 of PMS.

Comment by Reviewer 3:

  1. The SEM images (Figure. 6) must be in same scale bars or otherwise mention the scale bars in the Figure caption.

Response from authors:

The SEM images in Figure 6 were corrected according Reviewer comment. SEM images were presented in the same scale.

Figure 6. FE-SEM images of PANI/AuNPs(AuCl4-)-GOx composite (A) and Ppy/AuNPs(AuCl4-)-GOx composite (B) formed after 2 days of enzyme-assisted formation.

We will thank for positive feedback and valuable recommendations.

We hope after all these corrections our manuscript is suitable for publication.

Yours sincerely,
Arunas Ramanavicius

----------------------------------------------------------------
Prof. habil. dr. Arunas Ramanavicius

Head of Department of Physical Chemistry,

Faculty of Chemistry, Vilnius University,

Naugarduko 24, 03225 Vilnius 6, Lithuania; e-mail: [email protected]
